# Factors Contributing to Delayed Return to Work among French Healthcare Professionals Afflicted by COVID-19 at a Hospital in the Rhône-Alpes Region, 2021

**DOI:** 10.3390/ijerph20216979

**Published:** 2023-10-26

**Authors:** David Monier, Paul Bonjean, Pierre Carcasset, Martine Moulin, Bruno Pozzetto, Elisabeth Botelho-Nevers, Luc Fontana, Carole Pelissier

**Affiliations:** 1Occupational Health Service, University Hospital Center of Saint-Etienne, 42055 Saint-Etienne, France; d.monier@slst.fr (D.M.); pierre.carcasset@chu-st-etienne.fr (P.C.); martine.moulin@chu-st-etienne.fr (M.M.); luc.fontana@univ-st-etienne.fr (L.F.); 2Public Health Service, University Hospital Center of Saint-Etienne, 42005 Saint-Etienne, France; paul.bonjean@ch-roanne.fr; 3Department of Infectious Agents and Hygiene, University-Hospital of Saint-Etienne, 42055 Saint-Etienne, France; bruno.pozzetto@chu-st-etienne.fr; 4Infectious Diseases Department, University Hospital of Saint-Etienne, 42055 Saint-Etienne, France; elisabeth.botelho-nevers@chu-st-etienne.fr; 5University Lyon 1 University de St Etienne University Gustave Eiffel—UMRESTTE UMR_T9405, 42005 Saint-Etienne, France

**Keywords:** delayed return to work, COVID-19, hospital staff, persistent symptoms

## Abstract

COVID-19 is an emerging disease whose impact on the return to work of hospital staff is not yet known. This study was aimed at evaluating the prevalence of delayed return to work associated with medical, personal, and professional factors in hospital staff who tested positive for COVID-19 during the second epidemic wave. A descriptive, analytical observational study was conducted. The source population consisted of all staff of a French University Hospital Center who had an RT-PCR test or an antigenic test positive for SARS-CoV-2 during the period from 6 September to 30 November 2020. A delayed return to work was defined as a return to work after a period of at least 8 days of eviction, whereas before the eviction period decided by the French government was 14 days. Data collection was carried out through an anonymous online self-questionnaire. The participation rate was 43% (216 participants out of 502 eligible subjects). Moreover, 40% of the staff had a delayed return to work, and 24% of them reported a delayed return to work due to persistent asthenia. Delayed return to work was significantly associated with age, fear of returning to work, and persistent asthenia, but the number of symptoms lasting more than 7 days was the only factor that remained significantly associated after multivariate analysis. From this study, it appears that interest in identifying the number of persistent symptoms as a possible indicator of delayed work emerges. Moreover, persistent asthenia should be given special attention by practitioners to detect a possible long COVID.

## 1. Introduction

Coronavirus disease 2019 (Covid-19) is caused by the SARS-CoV-2 virus, which is spread throughout the population mainly by close contact with an infected person through the projection of contaminated droplets.

France faced a strong increase in COVID-19 incidence during fall 2020, with a peak at 899 per 100,000 persons in the Rhone Alpes region. During the second wave of the pandemic, the number of hospitalized COVID-19 patients increased, with an increase in the number of health-care workers (HCW) infected by COVID-19 [1].

Since 2020, the COVID-19 pandemic has had an impact on workers physical and mental health [2]. Aymerich et al. underlined in a systematic review and a meta-analysis that HCWs exposed to COVID-19 were found to have a significant prevalence of mental health concerns (depressive symptoms, anxiety features, and post-traumatic symptoms) [3]. The direct impact was having to manage the contagion risk in the workplace and returning to work post-COVID-19 [4].

In hospitals, there is a considerable risk of contracting COVID-19 infection among health care workers (HCWs), with a high incidence of mild to moderate forms in this population [5]. Hospital personnel are exposed to the risk of infectious disease contamination both through their care activities with potentially infected patients and through contact with their relatives [6].

In France, the national “test, alert, protect” strategy was implemented in September 2020 to curb the spread of the virus. This strategy is based on detecting symptomatic subjects with COVID-19 and contact subjects. Among the diagnostic tests to confirm COVID, the reverse transcription reaction followed by a real-time quantitative chain polymerization reaction (RT_PCR) and rapid diagnostic testing based on specific SARS-CoV-2 antigen detection in the early phase of infectious manifestations have been used [7].

The main symptoms of COVID-19 infection are fever, respiratory signs (coughing, shortness of breath), headaches, myalgia, arthralgia, unusual tiredness, anosmia, dysgueusia, and diarrhea [8].

At the Defense Council meeting on 11 September 2020, the government decided to shorten the isolation period for COVID-19 patients and contact cases from 14 to 7 days, in accordance with the opinion of the Scientific Council issued on 3 September 2020.

Is the period of isolation sufficient to allow the necessary recovery to return to work?

Gualano et al. conducted a systematic review to evaluate the impact of lasting COVID-19 symptoms or disability on the working population upon their return to employment [4]. Chopra et al. reported that 60% of patients hospitalized for COVID-19 and employed full- or part-time before COVID-19 hospitalization had returned to work after two months [9]. Aben et al. conducted a study among Dutch employees who reported being sick due to COVID-19 (N = 30,396). The median time-to-RTW after COVID-19 was 10 days [10]. Ganz-lord et al. included health care workers (HCWs) who called the Occupational Health Service to report COVID-19 symptoms between 1 March and 12 June 2020 [11]. They showed that the median time from symptom onset until return to work for HCWs who did not require hospitalization was 15 days.

The objective of this study was to assess the prevalence of delayed return to work and identify associated medical, personal, and occupational factors in hospital staff who tested positive for COVID-19 during the second wave of the epidemic in France.

## 2. Materials and Methods

The study design consisted of a cross-sectional questionnaire survey.

### 2.1. Target Population

The target population was employees infected with the SARS-CoV-2 virus during the second epidemic wave who worked in the University Hospital of Saint-Etienne, located in the Rhône-Alpes region.

### 2.2. Study Sample

During the period from 6 September to 30 November 2020 (period of the second epi-demic wave in France), 562 hospital staff (135 men and 427 women) among 7754 employees of the University Hospital of Saint-Etienne exhibited a positive nasopharyngeal swab result (RT-PCR or antigenic test).

The hospital staff were invited to respond voluntarily to a self-administered online survey on 4 March 2021 via email.

Inclusion criteria for the eligible subjects:-To be over 18 years old at the time of infection with SARS-CoV-2.-To be employed by the University Hospital of Saint-Etienne.-To have had an RT-PCR test or a positive antigen test for SARS-CoV-2 during the period from 6 September to 30 November 2020.

Exclusion Criteria for the Eligible Subjects:-Be off work or on leave at the time of inclusion in the study.-Absence of telephone contact details, e-mail addresses, or invalid e-mails

Of the 562 eligible employees, 60 were excluded from participation in the study due to the absence of telephone contact details, e-mail addresses, or invalid e-mails.

Participants received clear and comprehensible information on the study objectives and procedure and were free to decline participation. The review board approval (IRBN082021/CHUSTE, 14 January 2021) was obtained before starting the study. If they agreed to participate in the study, they completed an anonymous online questionnaire via the LimeSurvey application.

### 2.3. Measurements

We developed a self-reported questionnaire to collect data on demographic, occupational, and medical characteristics. The self-administration time was approximately 10 min. The data was collected between 4 March and 15 March 2021 by means of an online questionnaire.

The main variable is the period of absence from work. A delayed return to work was defined as a period of absence from the workstation (time off work, leave, or teleworking) of more than 7 days. This information was collected in the questionnaire through the question “How many days were you absent from work, or teleworking, due to your COVID-19 infection?”. It was therefore collected in quantitative form and on a declarative basis. A delayed return to work was considered when the answer to the above question was greater than seven days. This cut-off was selected based on the French Defense Council decision at the time of the study, which was to isolate infected people for a minimum of 7 days.

The anonymous self-administered questionnaire covered 3 areas with 16 single- and 4 multiple-choice questions:-Personal: gender, age, and marital status (3 questions).-Occupational: occupational category, hospital unit, pace and hours of work, changes to the workplace, and concern about returning to work (8 questions).-Medical: comorbidity, SARS-CoV-2 infection, COVID-19 symptoms, duration of each symptom, hospitalization related to COVID-19 (9 questions).

Persistent symptoms of COVID-19 were investigated using the following question:

Have you had any persistent symptoms of COVID-19 that have delayed (or are still delaying) your return to work?

-Yes, persistent fatigue or malaise.-Yes, persistent fever.-Yes, persistent headache.-Yes, persistent muscle or joint pain.-Yes, persistent coughing or breathing difficulties.-Yes, persistent diarrhea or stomach ache.-Yes, persistent loss of taste or smell.-Yes, persistent anxiety or anguish.-No, persistent symptoms.

Complications from COVID-19 were investigated using the following question: Have you experienced any complications from COVID-19 that have delayed (or are still delaying) your return to work? 12 suggestions were made to participants to describe complications: respiratory, cardiovascular, psychiatric, cognitive, neurological, rheumatologic, renal, hepatic, digestive, endocrine, obstetrical, or no complications.

### 2.4. Analysis

The minimum sample size calculation was based on several assumptions. The proportion of workers presenting a delayed return was expected to be around 40%, according to the literature. To achieve a study power of 80%, it was estimated that a sample size of 288 participants would be required to detect a significant difference with an odds ratio of 2 and an alpha risk of 5%. A descriptive analysis was made of the sample’s sociodemographic, occupational, and medical characteristics. A joinpoint regression was also conducted to identify trends in several symptom frequencies during the first thirty days. After observing graphical representations of these evolutions, a model with one joinpoint was realized for each symptom, and the slopes were calculated before and after the joinpoint. Univariate analysis was performed to identify factors associated with delayed return to work using Wilcoxon-Mann-Whitney and Chi^2^ tests. Variables associated with delayed return to work with *p* < 0.2 were included in a logistic regression model to investigate their multivariate relationships with the outcome. All tests were 2-sided, at 5% alpha risk. Statistical analyses used R software, version 4.0.0.

## 3. Results

### 3.1. Sociodemographic, Occupational, and Medical Characteristics (Figure 1)

The participation rate was 43% (216 subjects). The gender ratio is 0.32. The mean age was 36.4 years (σ = 11.6) compared to 38.2 years (σ = 11.7) in the source population. More than three-quarters of participants were healthcare workers, more than half were paramedical professions (53% compared with 26% medical professions and 21% non-healthcare staff), and they worked more than 36 h a week (52.2%).

79 subjects (40%) exhibited a delayed return to work. The average duration of absence from the workplace (including telework) was 10 days (σ = 7.6; median, 7 days).

Symptoms persisting for more than 7 days were found in 34.5% of cases. As shown in Figure 1, the symptoms that persist longest are fatigue, dyspnea, and anosmia/dysgueusia. The main persistent symptom was fatigue (24%).

The main complications were cognitive (6%), followed by respiratory, cardiovascular, psychiatric, rheumatologic, and digestive (1% each). No neurological, renal, hepatic, endocrine, or obstetric complications were reported. No respondent was hospitalized, and none died.

The results of the joinpoint regression analysis concerning symptoms evolution during the first thirty days are presented in Appendix A. Symptom frequencies of fever, headache, myarthralgia, and digestive disorders strongly declined until reaching values near zero during the first ten days (Appendix A). Frequencies of asthenia, respiratory disorders, anxiety, and anosmia declined until reaching joinpoints situated around twenty days, and after these, joinpoints stayed above zero at least until the thirtieth day (Appendix A).

**Figure 1 ijerph-20-06979-f001:**
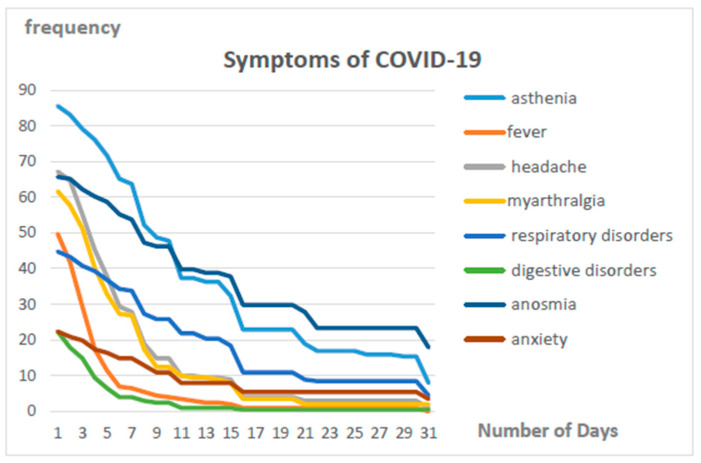
Trend in the duration of symptoms for COVID-19.

### 3.2. Relations between Delayed Return to Work and Occupational and Medical Factors on Univariate Analysis (Table 1)

In univariate analysis, the medical and socio-occupational factors significantly associated with a delayed return to work were:-age [OR = 1.03 (1.01–1.06)],-number of symptoms lasting more than 7 days [OR = 2.25 (1.77–2.96)],-concern of returning to work [OR = 2.58 (1.37–4.92)], and-fatigue for more than 7 days [OR = 7.11 (3.74–14.09)].

There was no significant association between time to return to work and gender, occupational category, presence of a risk factor (comorbidity, disability, or chronic illness), or occurrence of medical complications.

**Table 1 ijerph-20-06979-t001:** Medical and socio-occupational factors associated with delayed return to work.

Variables	Modalities	Delayed Return to Work			
		No (n = 115)	Yes (n = 79)	OR (95% CI)	Adjusted OR (95% CI)
**Quantitative variables**		**Median (Q1–Q3)**	**Median (Q1–Q3)**		
- age	-	33 (25–42)	39 (29–48)	1.03 (1.01–1.06) *	1.02 (0.99–1.05)
- number of symptoms > 7 days	-	1 (0–2)	3 (2–4)	2.25 (1.77–2.96) ***	2.03 (1.45–2.92) ***
**Categorical variables**		N (%)	N (%)		
- gender	Female	93 (59.6)	63 (40.4)	1	-
	Male	22 (57.9)	16 (42.1)	1.07 (0.52–2.19)	-
- comorbidity, disability, or chronic disease	No	93 (61.6)	58 (38.4)	1	-
	Yes	22 (51.2)	21 (48.8)	1.53 (0.77–3.04)	-
- occupational category	Medical	34 (68.0)	16 (32.0)	1	-
	Paramedical	61 (58.1)	44 (41.9)	1.53 (0.76–3.17)	-
	Non-caregiver	20 (51.3)	19 (48.7)	2.02 (0.86–4.85)	-
- medical complications	No	108 (61.0)	69 (39.0)	1 ^†^	1
	Yes	7 (41.2)	10 (58.8)	2.24 (0.82–6.42)	0.45 (0.10–1.94)
- concern of returning to work	No	91 (65.9)	47 (34.1)	1 **	1
	Yes	24 (42.9)	32 (57.1)	2.58 (1.37–4.92)	1.18 (0.54–2.57)
- fatigue > 7 days	No	76 (81.7)	17 (18.3)	1 ***	1
	Yes	39 (38.6)	62 (61.4)	7.11 (3.74–14.09)	1.60 (0.64–4.01)

Q1: 1st quartile; Q2: 2nd quartile; OR: Odds Ratio; CI: Confidence interval. ^†^ *p*-value < 0.2; * *p*-value < 0.05; ** *p*-value < 0.01; *** *p*-value < 0.001.

### 3.3. Relations between Delayed Return to Work and Occupational and Medical Factors on Multivariate Analysis (Table 1)

In multivariate analysis, the number of symptoms lasting more than 7 days [OR = 2.03 (1.45–2.92)] was the only factor that remained associated with a delayed return to work. This association was independent of the other variables included in the multivariate model.

## 4. Discussion

Our results regarding the prevalence of delayed returns to work are consistent with some literature data (40%). Gaber et al. conducted an observational survey to investigate the long-term impact of COVID-19 among HCWs. Of the 114 individuals infected with SARS-CoV-2 during the first wave of the COVID-19 crisis, 102 HCWs had a period of less than 14 days of sick leave [12]. A retrospective cohort study included 155 Italian patients who completed follow-up within a year of testing positive for COVID-19. Buonsenso et al. showed that 9.0% of patients reported not feeling fully recovered at follow-up, and 13.7% reported a change in their job status after COVID-19 [13]. According to Breugnon et al., the median time to recovery was 9 days after the onset of symptoms [14].

Despite the limited size of the effectives, the participation rate was satisfactory to control for the potential effect of confounding factors; an adjustment by multivariate analysis was performed.

In our study, there was no significant association between time to return to work and gender, occupational category, presence of a risk factor (comorbidity, disability, or chronic illness), or occurrence of medical complications.

Aben et al. conducted a study among Dutch employees who reported being sick due to COVID-19 (N = 30,396) [10]. They underlined that the main predictors contributing to a later return to work were older age, female sex, belonging to a risk group, and the symptoms of shortness of breath and fatigue.

Jacobsen et al. conducted a retrospective study among 7466 patients sick due to COVID-19 to explore their return to work after COVID-19 [15]. They showed that female sex, older age, and comorbidity were associated with a lower chance of returning to work. The low sample size and the potential bias of the healthy and young workers in our study are factors that may have contributed to the lack of evidence of a significant association between time to return to work and these factors.

Our study highlights that the delayed return to work was associated with a number of symptoms lasting more than 7 days after multivariate analysis and in a non-independent way with age, concern of returning to work, and persistent fatigue. Keech et al. showed a positive association between the number of reported symptoms and the number of days off work in the context of influenza infection [16]. The retrospective cohort study by Ganz-Lord et al. of 1698 healthcare workers with COVID-19 infection from March to June 2020 found that employees who did not require hospitalization returned to work at a median of 15 days from symptom onset; dyspnea, fever, sore throat, and diarrhea were significantly associated with a delayed return to work [11].

Our study highlights that fatigue, anosmia, dysgueusia, and dyspnea are the most common lasting symptoms after COVID-19. These findings are consistent with those of Mandic-Rajcevi et al., who showed in 172 SARS-CoV-2-positive healthcare workers a gradual reduction of most symptoms in the second week, with persisting loss of smell and loss of taste in around 30% of them [17].

According to the recommendations of the French High Council for Public Health (HCSP), hospital personnel were able to resume their work as soon as the fever and possible dyspnea disappeared on the eighth day after the onset of symptoms, keeping a surgical mask on for seven days (fourteen days if they are immunocompromised). However, fatigue was the most frequent persistent symptom, which could delay the return to work. Our results are consistent with those of the literature, which showed that fatigue is the most frequent symptom after SARS-CoV-2 infections [18]. Lulli et al. identified that over half of healthcare workers infected with COVID-19 mentioned at least one symptom persisting, and fatigue was the most reported disorder [19]. According to a systematic review, fatigue affects more than 50% of patients [20].

According to Magnavita et al., more than two-thirds of workers who contracted SARS-CoV-2 in the early stages of the pandemic experienced symptoms that persisted for more than four weeks after the acute phase, such as anosmia/dysgueusia or fatigue [21].

Moreover, Chopra et al. conducted an observational study of 488 patients hospitalized for COVID-19 (195 were working prior to the COVID-19 infection) [9]. They found that of the 117 patients who returned to work, 30 had benefited from a modified workplace (reduced hours and/or modified duties upon return to work due to health).

An interest in identifying the number of persistent symptoms as a possible indicator of delayed work emerges. Persistent fatigue should be given special attention by practitioners to detect a possible long COVID [22].

Our study could be supplemented by a prospective follow-up of a cohort of workers infected with COVID-19 to assess the value of early detection of persistent symptoms such as fatigue and consultation with an occupational physician to adapt to the workplace.

## 5. Conclusions

This study shows the value of early identification of persistent symptoms of COVID-19, in particular fatigue, as an indicator of a delayed return to work.

Recent studies have highlighted persistent fatigue as one of the signs of COVID-Long [23,24]. These results underline the importance of strengthening communication between physicians and occupational physicians so as to improve the possibilities of adapting the workplace when returning to work.

Strategies promoting a return to work for those with post-COVID-19 conditions will need to be implemented [25]. Occupational practitioners could be included in the process as early as possible to make job accommodations for improving the work abilities of such workers [24].

## Data Availability

The data presented in this study are available on request from the corresponding author.

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
