# Peer review of "Factors Contributing to Delayed Return to Work among French Healthcare Professionals Afflicted by COVID-19 at a Hospital in the Rhône-Alpes Region, 2021"

_ijerph, 2023, doi:10.3390/ijerph20216979_

Round 1

Reviewer 1 Report

Comments and Suggestions for Authors

Dear authors,

Thank you for submitting the manuscript “Socio-occupational and Medical Factors Associated with De- 2 layed Return to Work in Hospital Staff Affected by COVID-19”. The results section especially is very well written and thought out, with figures and tables that engage the reader. While the study is interesting, there are some issues in the manuscript:

1.       The introduction is too brief and does not explain the issue appropriately.

a.      The impact of the COVID-19 pandemic on healthcare workers should be further explored in the introduction: which health consequences has COVID-19 led to for HCWs? And which mental health consequences? (see: Rossi et al. “Coping with burnout and the impact of the COVID-19 pandemic on workers' mental health: A systematic review).

b.      Contrarily of what you state in line 50, the issue of returning to work after suffering from COVID-19 has been vastly investigated in scientific literature, the average % and days should be mentioned (as an example, see: Gualano et al. “Returning to work and the impact of post COVID-19 condition: A systematic review).

2.       Methods:

a.      The institutional review board (IRB) or ethics committee approval must be stated in the methods section with the number of approval.

3.       In the discussion section:

a.      While the discussion is engaging, there are additional factors that should be considered and integrated into this section. For instance, a comparison with studies performed in other countries on returning to work after COVID-19 would be interesting: an Italian study reported that 19% of workers who suffered from COVID-19 did not feel recovered when resuming work, and highlighted an increase in fatigue and anxiety (Buonsenso et al. “Post-Acute COVID-19 Sequelae in a Working Population at One Year Follow-Up: A Wide Range of Impacts from an Italian Sample”).

4.       Minor revisions should be made to address some oversights (one of the authors’ name may be incorrect in the online form? There is a “ in the abstract after the word Delayed that I don’t believe is intentional? And so on).

Comments on the Quality of English Language

The English language requires minor revisions

Author Response

We thank the three reviewers for their suggestions of modifications which contribute to improve the article

 REVIEW-1

Dear authors,

Thank you for submitting the manuscript “Socio-occupational and Medical Factors Associated with De- 2 layed Return to Work in Hospital Staff Affected by COVID-19”. The results section especially is very well written and thought out, with figures and tables that engage the reader. While the study is interesting, there are some issues in the manuscript:

  1. The introduction is too brief and does not explain the issue appropriately.
  2. The impact of the COVID-19 pandemic on healthcare workers should be further explored in the introduction: which health consequences has COVID-19 led to for HCWs? And which mental health consequences? (see: Rossi et al. “Coping with burnout and the impact of the COVID-19 pandemic on workers' mental health: A systematic review).

 Response to reviewer 1: The authors modified the introduction to align with the reviewer’s guidelines.

 They added the following sentences page 1 lines 33-38:

Since 2020, the COVID-19 Pandemic has had an impact on workers physical and mental health(2).Aymerich et al. underlined in a systematic review and a meta-analysist, that HCWs exposed to COVID-19 were found to have a significant prevalence of mental health concerns (depressive symptoms, anxiety features, and post-traumatic symptoms)(3) .The direct impact was having to manage the contagion risk in the workplace and returning to work post COVID-19(4).

  1. Rossi MF, Gualano MR, Magnavita N, Moscato U, Santoro PE, Borrelli I. Coping with burnout and the impact of the COVID-19 pandemic on workers’ mental health: A systematic review. Front Psychiatry. 2023;14:1139260.
  2. Aymerich C, Pedruzo B, Pérez JL, Laborda M, Herrero J, Blanco J, et al. COVID-19 pandemic effects on health worker’s mental health: Systematic review and meta-analysis. European Psychiatry. janv 2022;65(1):e10.
  3. Gualano MR, Rossi MF, Borrelli I, Santoro PE, Amantea C, Daniele A, et al. Returning to work and the impact of post COVID-19 condition: A systematic review. Work. 2022;405‑13.

  1. Contrarily of what you state in line 50, the issue of returning to work after suffering from COVID-19 has been vastly investigated in scientific literature, the average % and days should be mentioned (as an example, see: Gualano et al. “Returning to work and the impact of post COVID-19 condition: A systematic review).

Response to reviewer 1: : The authors modified the introduction to align with the reviewer’s guidelines.

 They added the following sentences page 2 lines 57-61:

Gualano et al. conducted a systematic review to evaluate the impact of lasting COVID-19 symptoms or disability on the working population upon their return to employment(4). Chopra et al. reported that 60% of patients hospitalized to COVID-19 and  employed full- or part-time before COVID-19 hospitalization, had returned to work at two months (9).Aben et al. conducted a study among Dutch employees who reported sick due to covid-19 (N=30 396). Median time-to-RTW after COVID-19 was after 10 days (10). Ganz-lord et al  included Health Care Workers (HCWs) who called Occupational Health Service to report COVID-19 symptoms between March 1 and June 12, 2020 (11). They showed that the median time from symptom onset until return to work for HCWs who did not require hospitalization was 15 days.

  1. Gualano MR, Rossi MF, Borrelli I, Santoro PE, Amantea C, Daniele A, et al. Returning to work and the impact of post COVID-19 condition: A systematic review. Work. 2022;405‑13.
  2. Chopra V, Flanders SA, O’Malley M, Malani AN, Prescott HC. Sixty-Day Outcomes Among Patients Hospitalized With COVID-19. Ann Intern Med. 20 avr 2021;174(4):576‑8.
  3. Aben B, Kok RN, Wind A de. Return-to-work rates and predictors of absence duration after COVID-19 over the course of the pandemic. SCAND J WORK ENV HEA. 1 avr 2023;49(3):182‑92.
  4. Ganz-Lord FA, Segal KR, Rinke ML. COVID-19 symptoms, duration, and prevalence among healthcare workers in the New York metropolitan area. Infect Control Hosp Epidemiol. 1 août 2021;42(8):917‑23.

  1. Methods:
  2. The institutional review board (IRB) or ethics committee approval must be stated in the methods section with the number of approval.

Response to reviewer 1: The institutional review board (IRB) has been stated in the methods section with the number of approval.

Page 3 line 94 : « The review board approval (IRBN082021/CHUSTE, 14 January 2021)) was obtained before starting the study. »

  1. In the discussion section:
  2. While the discussion is engaging, there are additional factors that should be considered and integrated into this section. For instance, a comparison with studies performed in other countries on returning to work after COVID-19 would be interesting: an Italian study reported that 19% of workers who suffered from COVID-19 did not feel recovered when resuming work, and highlighted an increase in fatigue and anxiety (Buonsenso et al. “Post-Acute COVID-19 Sequelae in a Working Population at One Year Follow-Up: A Wide Range of Impacts from an Italian Sample”).

Response to reviewer 1: The authors modified the discussion in order to align with the reviewer’s guidelines. They added the following sentences.

Page 5 line 197 :

Gaber et al. conducted an observational survey to investigate long-term impact of COVID-19 among HCWs. Of the one hundred and fourteen infected with SARS-CoV-2 during the first wave of the COVID-19 crisis, one hundred and two HCWs had a period of less than 14 days sick leave (12). A retrospective cohort study included 155 Italian patients who completed follow-up within a year of testing positive for COVID-19.  Buonsenso et al. showed that 9.0% of patients reported not feeling fully recovered at follow-up, and 13.7% reported a change in their job status after COVID-19 (13).

  1. Buonsenso D, Gualano MR, Rossi MF, Valz Gris A, Sisti LG, Borrelli I, et al. Post-Acute COVID-19 Sequelae in a Working Population at One Year Follow-Up: A Wide Range of Impacts from an Italian Sample. Int j environ res public health (Online) [Internet]. 2022 [cité 18 sept 2023]; Disponible sur: https://www.ncbi.nlm.nih.gov/pmc/articles/PMC9518581
  2. Breugnon E, Thollot H, Fraissenon A, Saunier F, Labetoulle R, Pillet S, et al. COVID-19 outpatient management: Shorter time to recovery in Healthcare workers according to an electronic daily symptoms assessment. Infectious Diseases Now. 1 févr 2021;51(1):71‑6.

Page 6 line 249

“Moreover, Chopra et al. conducted an observational study of 488 patients hospitalized for COVID-19 (195 were working prior to COVID-19 infection)(9). They found that of the 117 patients who returned to work, 30 had benefited from a modified workplace (reduced hours and/or modified duties upon return to work due to health).”

  1. Chopra V, Flanders SA, O’Malley M, Malani AN, Prescott HC. Sixty-Day Outcomes Among Patients Hospitalized With COVID-19. Ann Intern Med. 20 avr 2021;174(4):576‑8.

  1. Minor revisions should be made to address some oversights (one of the authors’ name may be incorrect in the online form? There is a “ in the abstract after the word Delayed that I don’t believe is intentional? And so on).

Response to reviewer 1: The authors modified the abstract and the author’s name

Reviewer 2 Report

Comments and Suggestions for Authors

I would like to extend my sincere gratitude to the authors for generously sharing their manuscript with us. However, there are some issues that should be addressed before considering the manuscript suitable for publication.

Since the article is focused on a sample of healthcare workers in a hospital in the Rhône-Alpes region, I recommend modifying the manuscript's title to: 'Factors Contributing to Delayed Return to Work Among Healthcare Professionals Afflicted by COVID-19 at a Hospital in the Rhône-Alpes Region, 2021.'

The abstract should be adjusted to align with the journal's guidelines.

Please provide clarification regarding how you determined the dependent variable 'delayed return to work' and specify the source of this information.

Additionally, please explain how you arrived at the minimum sample size used for the study.

Lastly, kindly specify the study's inclusion timeframe.

Regarding Line 107, it states, 'More than half of the participants were healthcare professionals (53%, compared to 26% in medical professions and 21% in non-healthcare roles), and they worked more than 36 hours per week (52.2%). Please note that all individuals in this group are considered healthcare workers.'

For Figure 1, I recommend considering an alternative approach for trend measurement, such as a joint point analysis or a p-trend analysis, which may be more appropriate in this context.

Regards,

Comments on the Quality of English Language

The following examples illustrate grammatical errors, but this list is not exhaustive. I strongly encourage the authors to have the manuscript reviewed by a native English speaker

Lines  60 - 72

Grammatical error: "epi- 62" should be "epidemic wave in France),".

Grammatical error: "To Have had" should be "To have had

Lines 106 - 117

Suggested change: Consider adding parentheses to clarify the meaning of the percentages in the description of professions (e.g., "53% compared with 26% medical professions") to improve comprehension.

Suggested change: You may want to provide additional details to clarify certain points, such as specifying that "telework" was included in the duration of absence from work.

Suggested change: To enhance the flow and structure of the text, it could be helpful to rephrase some sentences.

Author Response

We thank the three reviewers for their suggestions of modifications, which contribute to improve the article

REVIEWER 2:

Since the article is focused on a sample of healthcare workers in a hospital in the Rhône-Alpes region, I recommend modifying the manuscript's title to: 'Factors Contributing to Delayed Return to Work Among Healthcare Professionals Afflicted by COVID-19 at a Hospital in the Rhône-Alpes Region, 2021.'

Response to reviewer 2:

P1 line 1:The authors modified the manuscript’s title : 'Factors Contributing to Delayed Return to Work Among French Healthcare Professionals Afflicted by COVID-19 at a Hospital in the Rhône-Alpes Region, 2021’

The abstract should be adjusted to align with the journal's guidelines.

Response to reviewer 2:

The authors adjusted the abstract in order to align it with the journal’s guidelines.

P1 line 7-22

COVID-19 is an emerging disease whose impact on the return to work of hospital staff is not yet known. This study was aimed to evaluate the prevalence of delayed return to work associated with medical, personal and professional factors in hospital staff tested positive for COVID-19 during the second epidemic wave. A descriptive, analytical observational study was conducted. The source population consisted of all staff of a French University Hospital Center who had an RT-PCR test or an antigenic test positive for SARS-CoV-2 during the period from September 6 to November 30, 2020. Delayed return to work was defined as a return to work after a period of at least 8 days of eviction, whereas before the eviction period decided by the French government was 14 days. Data collection was done by an anonymous online self-questionnaire. The participation rate was 43% (216 participants out of 502 eligible subjects). 40% of the staff had a delayed return to work and 24% of  them reported delayed return to work due to persistent asthenia. Delayed return to work was significantly associated with age, fear of returning to work, persistent asthenia but the number of symptoms lasting more than 7 days was the only factor that remains significantly associated after multivariate analysis. From this study, it appears the interest of identifying the number of persistent symptoms as a possible indicator of delayed to work emerges.  Moreover, persistent asthenia should be given special attention by practitioners to detect a possible long COVID.

Please provide clarification regarding how you determined the dependent variable 'delayed return to work' and specify the source of this information.

Response to reviewer 2:

The authors added a few sentences to explain how this information was  determined (page 3  line 103-111)

A delayed return to work was defined as a period of absence from the workstation (time off work, leave or teleworking) of more than 7 days. This information was collected in the questionnaire through the question "How many days were you absent from work, or teleworking, due to your COVID-19 infection?". It was therefore collected in quantitative form and on a declarative basis. A delayed return to work was considered when the answer to the above question was greater than seven days. This cut-off was selected based on the French Defense Council decision at the time of the study, which was to isolate infected people for a minimum of 7 days”.

Additionally, please explain how you arrived at the minimum sample size used for the study.

Response to reviewer 2:

The authors added a few sentences to explain how the number of subjects needed was estimated, page 3 line 137-141:

“The minimum sample size calculation was based on several assumptions. The proportion of workers presenting a delayed return was expected to be around 40% according to the literature. To achieve a study power of 80%, it was estimated that a sample size of 288 participants would be required to detect a significant difference with an odds ratio of 2 and an alpha risk of 5%.”

Lastly, kindly specify the study's inclusion timeframe.

Response to reviewer 2:

The authors have added the following sentence p3 line 101:

“The data was collected between 1 and 15 March 2021 by means of an online questionnaire”

Regarding Line 107, it states, 'More than half of the participants were healthcare professionals (53%, compared to 26% in medical professions and 21% in non-healthcare roles), and they worked more than 36 hours per week (52.2%). Please note that all individuals in this group are considered healthcare workers.'

Response to reviewer 2:

The authors have added the following sentence: p4 line 154” ”:

“More than three quarters of participants are healthcare workers, more than half of participants were paramedical professions (53% compared with 26% medical professions and 21% non-healthcare staff) and worked more than 36 hours a week (52.2%).”

For Figure 1, I recommend considering an alternative approach for trend measurement, such as a joint point analysis or a p-trend analysis, which may be more appropriate in this context.

Response to reviewer 2:

The authors have added the following sentences:

Page 3 line 142 in “Materials and Methods”

“A joinpoint regression was also conducted to identifiy trends in several symptoms frequencies during the thirty first days. After observing graphical representations of these evolutions, a model with one joinpoint was realized for each symptom and the slopes were calculated before and after the joinpoint”.

 Page4 line 167 in “Results”

“The results of the joinpoint regression analysis concerning symptoms evolution during the thirty first days are presented in supplementary materials. Symptoms frequencies of fever, headache, myarthralgia and digestive disorders strongly declined until reaching values near zero during the first ten days (Figure S1, S1bis and Table S1). Frequencies of asthenia, respiratory disorders, anxiety and anosmia declined until reaching joinpoints situated around twenty days, and after these joinpoint stayed above zero at least until the thirtieth day (Figure S1, S1bis and Table S1)”.

Comments on the Quality of English Language

The following examples illustrate grammatical errors, but this list is not exhaustive. I strongly encourage the authors to have the manuscript reviewed by a native English speaker

Lines  60 - 72

Grammatical error: "epi- 62" should be "epidemic wave in France),".

Grammatical error: "To Have had" should be "To have had

Response to reviewer 2:

The manuscript has been initially reviewed by a native English speaker.

The grammatical errors identified above have been corrected by the authors.

Reviewer 3 Report

Comments and Suggestions for Authors

Dear authors, thank you for the opportunity to read your interesting research. This topic is certainly relevant for the purposes of recovery work with medical workers.

The introduction contains an analysis of research on the stated topic, as well as the purpose of the study.

There is one remark: Line 50 - Few studies have explored return to work and sick leave after COVID 19 infection 50 [6] - data from only one study are given (although several have been written), while the authors do not describe what was studied in it, what is important for the purpose of this study. It is necessary to expand the description of such studies in the introduction.

The sample of the study and the procedure for collecting material are spelled out in detail.

The research method requires a more detailed description. It is necessary to specify in more detail the content of the questions, their number in each block, the rationale for the set of these particular factors for consideration. Because Perhaps the authors did not take into account all of them.

The results of the study are described in an understandable language and illustrated by a figure and a table.

The discussion section of the results also needs to be supplemented. It is also necessary to list the factors that did not show significant relationships, and describe your assumptions about this.

There are no study restrictions. Since this is a short report, the authors should indicate further study of the issue or further analysis of the data obtained.

The article breaks off, there is no conclusion at the end of the article. Should be added.

In connection with the above, the article needs to be improved.

Best regards, reviewer

Author Response

We thank the three reviewers for their suggestions of modifications, which contribute to improve the article

REVIEWER 3

The introduction contains an analysis of research on the stated topic, as well as the purpose of the study.

There is one remark: Line 50 - Few studies have explored return to work and sick leave after COVID 19 infection 50 [6] - data from only one study are given (although several have been written), while the authors do not describe what was studied in it, what is important for the purpose of this study. It is necessary to expand the description of such studies in the introduction.

Response to reviewer 3:

The authors added the following sentences page 2 line 57-66:

“Gualano et al. conducted a systematic review to evaluate the impact of lasting COVID-19 symptoms or disability on the working population upon their return to employment(4). Chopra et al. reported that 60% of patients hospitalized to COVID-19 and  employed full- or part-time before COVID-19 hospitalization, had returned to work at two months (9). Aben et al. conducted a study among Dutch employees who reported sick due to covid-19 (N=30 396). Median time-to-RTW after COVID-19 was after 10 days (10). Ganz-lord et al  included Health Care Workers (HCWs) who called Occupational Health Service to report COVID-19 symptoms between March 1 and June 12, 2020 (11). They showed that the median time from symptom onset until return to work for HCWs who did not require hospitalization was 15 days. “

  1. Gualano MR, Rossi MF, Borrelli I, Santoro PE, Amantea C, Daniele A, et al. Returning to work and the impact of post COVID-19 condition: A systematic review. Work. 2022;405‑13.
  2. Chopra V, Flanders SA, O’Malley M, Malani AN, Prescott HC. Sixty-Day Outcomes Among Patients Hospitalized With COVID-19. Ann Intern Med. 20 avr 2021;174(4):576‑8.
  3. Aben B, Kok RN, Wind A de. Return-to-work rates and predictors of absence duration after COVID-19 over the course of the pandemic. SCAND J WORK ENV HEA. 1 avr 2023;49(3):182‑92.
  4. Ganz-Lord FA, Segal KR, Rinke ML. COVID-19 symptoms, duration, and prevalence among healthcare workers in the New York metropolitan area. Infect Control Hosp Epidemiol. 1 août 2021;42(8):917‑23.

The sample of the study and the procedure for collecting material are spelled out in detail.

The research method requires a more detailed description. It is necessary to specify in more detail the content of the questions, their number in each block, the rationale for the set of these particular factors for consideration. Because Perhaps the authors did not take into account all of them.

Response to reviewer 3:

The authors gave more details to describe the content of the questionnaire page 3 line 112-135.

“The anonymous self-administered questionnaire covered 3 areas with 16 single- and 4 multiple-choice questions.

- Personal: gender, age, marital status (3 questions)

- Occupational: occupational category, hospital unit, pace and hours of work, changes to workplace, concern of returning to work (8 questions)

- Medical: comorbidity, SARS-CoV-2 infection, COVID-19 symptoms, duration of each symptom, Hospitalisation related to COVID-19 (9 questions)

Persistent symptoms of COVID-19 were investigated using the following question:

Have you had any persistent symptoms of COVID 19 that have delayed (or are still delaying) your return to work?

  • Yes persistent fatigue or malaise
  • Yes persistent fever
  • Yes persistent headache
  • Yes persistent muscle or joint pain
  • Yes persistent coughing or breathing difficulties
  • Yes persistent diarrhea or stomach ache
  • Yes persistent loss of taste or smell
  • Yes persistent anxiety or anguish
  • No persistent symptoms

Complications from COVID-19 were investigated using the following question: Have you experienced any complications from COVID 19 that have delayed (or are still delaying) your return to work? 12 suggestions were made to participants to describe complications: respiratory, cardiovascular, psychiatric, cognitive, neurological, rheumatologic, renal, hepatic, digestive, endocrine, obstetrical, no complications.”

The results of the study are described in an understandable language and illustrated by a figure and a table.

The discussion section of the results also needs to be supplemented. It is also necessary to list the factors that did not show significant relationships, and describe your assumptions about this.

Response to reviewer 3:

The authors added the following sentences page 6 line 209-221:

“In our study, there was no significant association between time to return to work and gender, occupational category, presence of a risk factor (comorbidity, disability or chronic illness) or occurrence of medical complications.

Aben et al. conducted a study among Dutch employees who reported sick due to covid-19 (N=30 396)(10). They underlined that the main predictors contributing to later return to work were older age, female sex, belonging to a risk group, and the symptoms shortness of breath and fatigue.

Jacobsen et al. conducted a retrospective study among 7466 patients sick due to COVID-19 to explore return to work after covid-19 (15). They showed that female sex, older age and comorbidity were associated with a lower chance of returning to work. The low size sample, the potential bias of the healthy and young worker in our study, are factors that may have contributed to the lack of evidence of a significant association between time to return to work and these factors.”

  1. Aben B, Kok RN, Wind A de. Return-to-work rates and predictors of absence duration after COVID-19 over the course of the pandemic. SCAND J WORK ENV HEA. 1 avr 2023;49(3):182‑92.
  2. Jacobsen PA, Andersen MP, Gislason G, Phelps M, Butt JH, Køber L, et al. Return to work after COVID-19 infection - A Danish nationwide registry study. Public Health. 14 janv 2022;203:116‑22.

There are no study restrictions. Since this is a short report, the authors should indicate further study of the issue or further analysis of the data obtained.

Response to reviewer 3:

The authors added the following sentences page 6 line 249:

“Moreover, Chopra et al. conducted an observational study of 488 patients hospitalized for COVID-19 (195 were working prior to COVID-19 infection)(9). They found that of the 117 patients who returned to work, 30 had benefited from a modified workplace (reduced hours and/or modified duties upon return to work due to health).”

  1. Chopra V, Flanders SA, O’Malley M, Malani AN, Prescott HC. Sixty-Day Outcomes Among Patients Hospitalized With COVID-19. Ann Intern Med. 20 avr 2021;174(4):576‑8.

The authors added the following sentence page 6 line 256:

“Our study could be supplemented by a prospective follow-up of a cohort of workers infected with COVID-19 to assess the value of early detection of persistent symptoms such as fatigue and consultation with an occupational physician to adapt the workplace.”

The article breaks off, there is no conclusion at the end of the article. Should be added.

Response to reviewer 3: The authors added a conclusion

“This study shows the value of early identification of persistent symptoms of COVID-19, in particular fatigue, as an indicator of delayed return to work.

 Recent studies have highlighted persistent fatigue as one of the signs of COVID-Long(24,25). These results underline the importance of strengthening communication between physicians and occupational physicians, so as to improve the possibilities of adapting workplace when returning to work.

Strategies promoting return to work for those with post-COVID-19 conditions will need to be implemented (26). Occupational practitioners could be include in the process as early as possible to job accommodations for improving work ability of  such workers (25). “

  1. Fernández-de-Las-Peñas C, Palacios-Ceña D, Gómez-Mayordomo V, Cuadrado ML, Florencio LL. Defining Post-COVID Symptoms (Post-Acute COVID, Long COVID, Persistent Post-COVID): An Integrative Classification. Int j environ res public health (Online) [Internet]. 2021 [cité 20 sept 2023]; Disponible sur: https://www.ncbi.nlm.nih.gov/pmc/articles/PMC7967389
  2. Clin B, Esquirol Y, Gehanno JF, Letheux C, Gonzalez M, Pairon JC, et al. Rôle des services de santé au travail dans le repérage et l’accompagnement des personnes concernées par des symptômes persistants suite à la Covid-19. Recommandations de la Société française de médecine du travail (SFMT) Role of occupational health services in identifying and supporting people affected by persistent symptoms following Covid-19. French Society of Occupational Medicine Guideline. Archives Des Maladies Professionnelles et De L’Environnement. 1 juill 2021;82(4):395‑400.
  3. Descatha A, Evanoff BA, Fadel M. Post-COVID condition or « long COVID », return-to work, and occupational health research. Scand J Work Environ Health. 1 avr 2023;49(3):165‑9.

Round 2

Reviewer 1 Report

Comments and Suggestions for Authors

The manuscript has improved, introduction is clearer and discussion more compelling.

Comments on the Quality of English Language

English language is fine.

Reviewer 2 Report

Comments and Suggestions for Authors

The authors have carefully considered the referees' comments.

Well done!

Reviewer 3 Report

Comments and Suggestions for Authors

Dear authors, thanks for the additions!

Currently, the article has been improved, all comments have been corrected.

Best wishes, reviewer